# Knowledge of Thai women in cervical cancer etiology and screening

Uraiwan Khomphaiboonkij[1☉]*, Nattapong Sreamsukcharoenchai[2‡], Supakorn Pitakkarnkul[1‡], Kristsanamon Rittiluechai[2‡], Siriwan Tangjitgamol[3,4☉]

1 Department of Gynecologic Oncology, National Cancer Institute, Bangkok, Thailand, 2 Department of Obstetrics and Gynecology, Phramongkutklao Hospital, Bangkok, Thailand, 3 Department of Obstetrics and Gynecology, Faculty of Medicine Vajira Hospital, Navamindradhiraj University, Bangkok, Thailand, 4 Obstetrics and Gynecology Section, MedPark Hospital, Bangkok, Thailand

☉ These authors contributed equally to this work.
‡ NS, SP and KR also contributed equally to this work.
* awan1985@hotmail.com

**Data Availability Statement:** All relevant data are within the paper and its Supporting Information files.

## Abstract

Knowledge about cervical cancer screening and Human papilloma virus (HPV) influence on their awareness to the cervical cancer screening program. Most previous studies found inadequate knowledge and attitude among healthy women affect the low rate of screening. This study aimed to assess knowledge of cervical cancer screening and HPV in women who had abnormal cervical cancer screening in Bangkok. Thai women, aged ≥ 18 years old, who had abnormal cervical cancer screening and scheduled to colposcopy clinics of 10 participating hospitals were invited to participate in this cross-sectional study. The participants were asked to complete a self-answer questionnaire (Thai language). The questionnaire composed of 3 parts: (I) demographic data, (II) knowledge about cervical cancer screening and (III) knowledge about HPV. Among 499 women who answered the questionnaires, 2 had missing demographic data. The mean age of the participants was 39.28 ± 11.36 years. 70% of them had experience of cervical cancer screening, with 22.7% had previous abnormal cytologic results. Out of 14 questions, the mean score of knowledge about cervical cancer screening was 10.04 ± 2.37. Only 26.9% had good knowledge about cervical cancer screening. Nearly 96% of woman did not know that screening should be done. After excluding 110 women who had never known about HPV, 25.2% had good knowledge about HPV. From multivariable analysis, only younger age (≤ 40 years) was associated with good knowledge of cervical cancer screening and HPV. In the conclusion, only 26.9% of women in this study had good knowledge regarding cervical cancer screening. Likewise, 20.1% of women who had ever heard about HPV has good knowledge about HPV. Providing information about cervical cancer screening and HPV should improve the women's knowledge and better adherence to the screening procedure.

## Introduction

In 2020, the World Health Organization (WHO) announced a global strategy to eliminate cervical cancer by Human Papilloma Virus (HPV) vaccination, screening and early detection of

**Funding:** The Winnergy Medical Public Company Limited provided all self-sampling Aptima Multitest Swab specimen collection kits. The funders had no role in study design, data collection and analysis, decision to publish, or preparation of the manuscript.

**Competing interests:** All of the authors have declared that no competing interests exist.

cervical cancer [1]. For screening, the target is that 70% of women should be screened with a high-performance test by 35 years of age and again by 45 years of age. In Thailand, the Royal Thai College of Obstetrics and Gynaecology (RTCOG) also released a practice guideline of cervical cancer screening for women aged 25–65 years through cervical cytology every 2 years, or HPV testing every 5 years for women aged ≥30 years [2].

The coverage of cervical cancer screening varies among countries and even within each country. The Thai National Cancer Institute, in 2012, reported that the screening coverage was 69% among the targeted women aged 35–60 years during the years 2005–2009 and only 28% among the women who should have the screening by guideline [3]. According to cervical cancer country profile, WHO reported in 2019 that the screening rate in Thailand was only 67% despite an extensive campaigns and health education to the community [4].

Studies from the US [5, 6] and one systematic review [7] reported many influencing factors of the screening rate, such as, ethnic, age, education, socioeconomic status, and health coverage. Lower screening rates have been found in women who were older, less-educated, lack of knowledge or a recognition of the importance of screening, certain family or cultural background, lower socioeconomic status or living in rural areas [5–7]. Other studies from various countries also reported the causes of not having cervical cancer screening: self-perception of good health or unavailability, shyness, anxiety about the abnormal results, problems of health access either the distance or the personnel, and reluctance to undergo pelvic examination or to reveal personal/ sexual history [8–10].

The major underlying reason for not undergoing cervical cancer screening is a lack of knowledge about its role in detection cervical lesions [6]. The knowledge as well as the attitude of women certainly influence on their awareness and adherence or negligence to the screening recommendation.

Few studies in Thailand explored the reasons why the women never had cervical cancer screening or inadequate screening [11–13]. Various reasons were reported: shyness, embarrassment, fear of pain, no awareness about the cause of cervical cancer and an importance of screening, unmarried, self-perception of being healthy or having no risk that the test deemed unnecessary, or anxiety about the abnormal results [11–13]. Although 83% were aware that cervical cancer could be detected by screening, however, only 26% had some degrees of knowledge regarding age to start, frequency and time to stop the screening [13].

Most previous studies assessed knowledge and attitude among healthy women. We do not know whether findings would be different among women who had history or were having abnormal cervical testing.

This study aimed to assess knowledge of cervical cancer screening and HPV in women who had abnormal cervical cancer screening in Bangkok, Thailand. We postulated that this group of women should have better knowledge than those who never had abnormal screening test at all.

## Materials and methods

This cross-sectional study was under the umbrella project of the Thai Gynecologic cancer Society to assess the clinical performance of self HPV testing. The study was collaboration among 10 tertiary hospitals in Bangkok, Thailand. The protocol was approved by Central Research Ethics Committee (CREC; COA-CREC082/2021). This study focused on knowledge of cervical cancer, its screening, and facts about HPV.

### Participants and eligibility criteria

The study was conducted from December 2021 until the end of June 2022. Inclusion criteria were women aged ≥ 18 years who had abnormal cervical cytology and/ or HPV testing and

who were appointed to undergo colposcopy in each participating hospital. The women who had prior treatment for invasive cervical cancer, metastatic cancer to cervix or uterus, were pregnant, or active vaginal bleeding were excluded. Fifty women from each 10 participating hospital were invited to the study.

## Study questionnaires

The questionnaire consisted of 3 parts. Part I included questions of socio-demographic data (age, level of education, parity, familial income, occupation, current marital status, sexual activity, history of HPV vaccination, and history of abnormal cervical cancer screening).

Part II comprised of 6 questions about the target women who should be screened, age to start, frequency, age or conditions to stop of screening, and methods of screening. A separate sheet of questions asking whether women were familiar with HPV. Only the participants who had ever heard of HPV would proceed to Part III which assessed knowledge about HPV regarding its type (oncogenic or non-oncogenic), mode of transmission and prevention.

Prior to the study, the questionnaire of part II and part III in Thai version were contemplated, discussed, and revised until consensus among 20 researchers who were all gynecologic oncologists (some were authors in other parallel studies). The experts rated each of 19 questions as either 'relevant' or 'not relevant'. The Item-level Content Validity Index (I-CVI) was calculated for each item, and the Content Validity Index (CVI) was calculated by taking the average proportion of 14 questions about knowledge about cervical cancer screening and 5 questions about HPV. The CVI were 1 for both parts of the questionnaire (S1 Table). The reliability was tested in 30 women with the same characteristic as the participants. The Cronbach' alpha coefficients for the reliability were 0.819 for knowledge about cervical cancer screening and 0.907 for knowledge about HPV.

## Data collection

On the date of scheduled colposcopy, the research assistant provided information of the study to the women who met inclusion criteria. The women who agreed to participate gave written informed consent and answered the questionnaire independently by themselves before proceeding to colposcopy.

Data collected for this study were socio-demographic data (Part I of the questionnaire), knowledge of cervical cancer screening (Part II), and knowledge of HPV (Part III).

## Statistical analysis

The answers of knowledge about cervical cancer screening and HPV were analyzed by each item before scoring categorization. Correct answer was scored as 1 whereas 'wrong and unsure answer' was scored 0.

Out of 14 questions of cervical cancer screening, women whose score were $\geq 10$ were arbitrarily classified as having 'good knowledge' or else were considered as having "not good knowledge". For knowledge of HPV, women whose answer was 'never heard about HPV' were excluded from the analysis of HPV knowledge.

Out of 5 questions, women whose score were $\geq 3$ were considered as having 'good knowledge of HPV' or else as 'not good knowledge'. Out of 19 questions, women whose score were $\geq 13$ were classified as having 'good overall knowledge' or else as 'not good overall knowledge'.

All data were collected in an electronic database and analyzed using SPSS V.26.0 (SPSS, Inc., Chicago, IL, USA). Descriptive statistics were used to analyze demographic data and were summarized as numbers with percentage, mean with standard deviation or median with

range. The association between each variable and knowledge score were analyzed using simple linear regression. Exploratory model was made in multivariable analysis including all variables potentially related to the knowledge score. The considered level of statistical significance was a p-value of 0.05 or less.

## Results

Out of 500 women who met inclusion criteria and were invited to the study, one declined to participate. Among 499 women who responded to the questionnaire about knowledge of cervical cancer and screening (part II) and knowledge about HPV (part II), two did not provide personal information. So demographic data were available in only 497 women (Table 1). The mean age of 497 participants was 39.28 years (SD ± 11.36). More than half of the participants were aged < 40 years old. Approximately 70% had a bachelor's degree or higher and 65.5% had higher monthly family income than the national average household income. Nearly 88% were unvaccinated against HPV whereas 72% were still sexually active. Among 70.0% who had ever had cervical cytology testing, 22.7% had history of abnormal cervical cytology. On the other hand, out of 21.7% who ever had HPV testing—15.3% had HR-HPV detected.

The indications for colposcopy were abnormal cervical cytology with presence of HR-HPV (155; 31.2%), abnormal cytology with negative or unknown of HR-HP (274; 55.1%), presence of only HR-HPV (65; 13.1%), or only history of CIN (3; 0.6%) (Table 2).

According to each question about cervical cancer screening (Table 3), 95.4% of 499 women misunderstood that all women should be screened for cervical cancer (Q 1). Approximately 55% to 70% knew when to start and frequency of screening (Q 2–4). However, only 17.4% and 36.3%were aware that the screening was not required further-—after hysterectomy for non-cervical cancer (Q 5) or after 65 years with history of consecutively normal previous testing (Q 6).

For the methods of screening (Q 7–14), 79% of participants were aware that there were many screening methods (Q 7). However, only 7.2% had knowledge that cervical screening could be done at the time of pelvic examination (Q 8). The participants were familiar with cervical cytologic testing (Q 10; 93.8%), HPV testing (Q 11; 89.2%), co-testing (Q 12; 92.4%) but not with visual inspection after acetic acid (VIA) which was known in only 31.7% (Q 9). Furthermore, 57.7% and 65.3% of participants had misconception that cervical cancer screening could be done by imaging (Q 13) and blood test for tumor markers (Q 14), respectively.

For knowledge of HPV, 110 participants (20.0%) had never heard about HPV. Among the remaining 389 participants, majority (92.5%) knew that HPV infection is preventable and 85.1% were aware that it is preventable by vaccination. Although majority (90.5%) knew that cervical cancer is due to oncogenic HPV, only approximately half knew that it can be prevented by avoiding sexual activity or venereal wart is due to non-oncogenic HPV.

We assessed knowledge of the participants by score (Table 4). The mean scores of participants for knowledge of cervical cancer screening (N = 499) and of HPV (N = 389) were 10.04 ± 2.37 (out of 14 questions) and 3.53 ± 1.11 (out of 5 questions), respectively. By categorization, 26.9% and 20.1% had good knowledge of cervical cancer screening and HPV respectively. The mean sum score of knowledge about cervical cancer screening and HPV were 13.72 ± 2.87 (out of 19 questions), with only 25.2% had good overall knowledge.

After excluding 2 women who did not complete socio-economic characteristic features and 110 women who had never heard about HPV, the association of knowledge and demographic data were studied in 389 participants. By univariable analysis, factors which were significantly associated with higher level of knowledge was younger age (≤ 40 year), single marital status, and never had previous screening test. Only younger age (≤ 40 year) remained significant by multivariable analyses (Table 5).

**Table 1. Socio-economic characteristic (N = 497).**

| Characteristics | N | Percent |
|---|---|---|
| Age group, mean age ± SD (years) | 39.28 ± 11.36 | |
| ≤40 | 280 | 56.3 |
| 41–60 | 191 | 38.4 |
| >60 | 26 | 5.2 |
| Marital status | | |
| Single | 128 | 25.8 |
| Married | 335 | 67.4 |
| Separate/divorces | 34 | 6.8 |
| Education level | | |
| Up to primary level | 42 | 8.5 |
| High school/ Diploma | 102 | 20.5 |
| Bachelor's degree | 286 | 57.5 |
| Master's degree and higher | 67 | 13.5 |
| Family monthly income (USD)* | | |
| ≤ 672 (<24k) | 171 | 34.4 |
| 673–1570 (24-50k) | 202 | 40.6 |
| > 1570 (>50k) | 124 | 24.9 |
| Occupation | | |
| Unemployed/ Student/ Housewife | 94 | 18.9 |
| Employee | 166 | 33.4 |
| Personal business | 86 | 17.3 |
| Government officer | 141 | 28.4 |
| Others | 10 | 2 |
| Sexual activity | | |
| Never | 6 | 1.2 |
| Ever, not active for 1 year | 133 | 26.8 |
| Still active within the past year | 358 | 72 |
| Parity, median (range) | 1 | (0–2) |
| Contraception | | |
| Never | 167 | 33.6 |
| Ever use | 330 | 66.4 |
| HPV vaccination | | |
| No | 436 | 87.7 |
| Yes | 61 | 12.3 |
| Bivalent | 10 | 2 |
| Quadrivalent | 40 | 8 |
| Nonavalent | 4 | 0.8 |
| Unknown | 7 | 1.4 |
| Previous cervical cancer cytology | | |
| Never had screening | 149 | 30 |
| Normal | 235 | 47.3 |
| Abnormal | 113 | 22.7 |
| Previous HPV testing | | |
| Negative | 32 | 6.4 |
| HPV 16,18 | 36 | 7.3 |
| Other HR-HPV | 40 | 8 |

(*Continued*)

**Table 1.** (Continued)

| Characteristics | N | Percent |
|---|---|---|
| Not done/ not available | 389 | 78.3 |

*Average familial income in Thailand 2,672 Baht/day = 672 USD per month (1 USD = 30 Baht).

## Discussion

This study found that only 25% of our participants had good knowledge about cervical cancer screening and HPV. This was close to 26% in one previous report from Bangkok, Thailand [13] despite different inclusion criteria. All of our participants had abnormal cervical testing (nearly 23% also had prior abnormal test) whereas all of their participants were healthy women without history of abnormal screening. The low level of knowledge in both studies were found despite the women in both studies were well-educated. Another study from a developed country (Sweden) also reported their participants who had high-grade CIN (cervical intraepithelial neoplasia) had lower specific knowledge about cervical cancer [14]. These findings tended to indicate that specific education of cervical cancer among the general population or even those with abnormal cervical testing or lesions were still inadequate. Our statement was supported from the finding of another study in Bangkok which found 47% of their participants had a high level of knowledge which was likely due to their participants being the hospital personnel [12]. Various levels of knowledge among women with diverse backgrounds emphasize that specific education about cervical cancer should be broadened to the population as much as possible to improve public knowledge.

When we explored each question of cervical cancer screening, as high as 79% of our participants were aware that there were many screening methods and approximately 90% knew the conventional methods (cytology) of screening. These may lie on the fact that there has been a national campaign by the Department of Medical Services, Ministry of Public Health (MOPH) that all women aged 30–60 years to have cervical cancer screening [15]. Nevertheless, we still found that 30% of our participants had never been screened at all. Unfortunately, our study did not explore the reasons for the non-adherence to screening program. Previous study in Thailand reported many reasons for not having a screening e.g. shyness, no symptoms and perception of being healthy, and etc. [11–13]. Among these barriers, lack of knowledge or misconception about screening might be the main underlying reasons. As found in our study that only 7.2% knew that these testing should be done with pelvic examination. Basic steps in obtaining the cervical sample may not be completely understood by the women.

**Table 2. Indication for colposcopy (N = 497).**

| Indication for colposcopy | N | Percent |
|---|---|---|
| Abnormal cytology with presence of HR-HPV | 155 | 31.2 |
| Positive HPV 16 or 18 | 51 | 10.3 |
| Positive other HR HPV | 104 | 20.9 |
| Abnormal cytology with negative or unknown of HR-HPV | 274 | 55.1 |
| HPV negative | 11 | 2.2 |
| Not done | 263 | 52.9 |
| Presence of HR-HPV only | 65 | 13.1 |
| HPV 16 or 18 | 51 | 10.3 |
| Other HR-HPV | 14 | 2.8 |
| No data of cytology or HR-HPV | 3 | 0.6 |

**Table 3. Questions and answers about knowledge of cervical cancer screening and HPV (N = 499).**

| Knowledge | Correct | | Wrong | | Unsure | |
|---|---|---|---|---|---|---|
| | n | (%) | n | (%) | n | (%) |
| **Knowledge about cervical cancer screening (N = 499)** | | | | | | |
| 1. All women should be screening. | 20 | -4 | 476 | -95.4 | 3 | -0.6 |
| 2. Screening should begin at age 25. | 345 | -69.1 | 144 | -28.9 | 10 | -2 |
| 3. Women who have never had sexual intercourse may begin the screening at age 30-year-old. | 273 | -54.7 | 204 | -40.9 | 22 | -4.4 |
| 4. If the results are normal, re-screening is done every 2–3 years or 5 years, depending on the screening method. | 351 | -70.3 | 132 | -26.5 | 16 | -3.2 |
| 5. Women who have their uterus removed along with the ovaries due to ovarian cancer still need continual screening. | 87 | -17.4 | 386 | -77.4 | 26 | -5.2 |
| 6. Women over 65 do not need further screening after consecutive normal screening results. | 181 | -36.3 | 287 | -57.5 | 31 | -6.2 |
| 7. There are several methods used for screening. | 394 | -79 | 84 | -16.8 | 21 | -4.2 |
| 8. Cervical cancer screening is recommended along with an annual pelvic examination | 36 | -7.2 | 461 | -92.4 | 2 | -0.4 |
| 9. Screening can be done by naked eye examination by a doctor/health worker after applying vinegar on the cervix. | 158 | -31.7 | 301 | -60.3 | 40 | -8 |
| 10. Screening can be done by cervical cytologic examination. | 468 | -93.8 | 24 | -4.8 | 7 | -1.4 |
| 11. Screening can be done by HPV testing. | 445 | -89.2 | 41 | -8.2 | 13 | -2.6 |
| 12. Screening can be done by cervical cytologic examination and HPV testing | 461 | -92.4 | 25 | -5 | 13 | -2.6 |
| 13. Screening can be done by ultrasound and computer tomography. | 185 | -37.1 | 288 | -57.7 | 26 | -5.2 |
| 14. Screening can be done by testing the blood for cancer markers. | 148 | -29.7 | 326 | -65.3 | 25 | -5 |
| Knowledge about HPV (N = 389*) | | | | | | |
| 15. HPV infection is preventable | 360 | -92.5 | 24 | -6.2 | 5 | -1.3 |
| 16. HPV infection can be prevented by avoiding sexual activity | 193 | -49.6 | 180 | -46.3 | 16 | -4.1 |
| 17. HPV infection can be prevented by vaccination | 331 | -85.1 | 49 | -12.6 | 9 | -2.3 |
| 18. Venereal wart is due to non-oncogenic types of HPV | 196 | -50.4 | 165 | -42.4 | 28 | -7.2 |
| 19. Cervical cancer is due to oncogenic types of HPV | 352 | -90.5 | 23 | -5.9 | 14 | -3.6 |

*Excluding 110 participants who had never heard about HPV.

Although one pioneer study in Thailand reported the role of VIA for cancer screening in a 'see and treat' approach in areas with inadequate health professional [16], only 32% were familiar with this technique. This was probably because VIA is generally promoted in low-resource

**Table 4. Knowledge score of cervical cancer screening and HPV.**

| Knowledge | N | Percent |
|---|---|---|
| Knowledge about cervical cancer screening (N = 499) | | |
| Knowledge score, Mean ± SD | | 10.04 ± 2.37 |
| Knowledge level, n (%) | | |
| Good (10–14 score) | 134 | -26.9 |
| Not good (0–9 score) | 365 | -73.1 |
| Knowledge about Human Papilloma Virus* (N = 389) | | |
| Knowledge score, Mean ± SD | | 3.53 ± 1.11 |
| Knowledge level, n (%) | | |
| Good (3–5 score) | 78 | -20.1 |
| Not good (0–2 score) | 311 | -79.9 |
| Knowledge total score*(N = 389) | | |
| Knowledge score, Mean ± SD | | 13.72 ± 2.87 |
| Knowledge level, n (%) | | |
| Good (13–19 score) | 98 | -25.2 |
| Not good (0–12 score) | 291 | -74.8 |

**Table 5. Characteristic features of the participants in association with knowledge score about cervical cancer screening and HPV.**

| Factors | Univariable analysis | | | Multivariable analysis | | |
|---|---|---|---|---|---|---|
| | Coef.[1] | 95%CI | p-value | Coef.[2] | 95%CI | p-value |
| Age | | | | | | |
| > 40 | 0 | Reference | | 0 | Reference | |
| ≤ 40 | 0.509 | (0.290, 0.728) | <0.001 | 0.502 | (0.246, 0.758) | <0.001 |
| Education level | | | | | | |
| Low to moderate | 0 | Reference | | 0 | Reference | |
| High | -0.027 | (-0.282, 0.228) | 0.836 | -0.192 | (-0.484, 0.100) | 0.198 |
| Familial income | | | | | | |
| Below average | 0 | Reference | | 0 | Reference | |
| Above average | -0.13 | (-0.367, 0.106) | 0.279 | -0.104 | (-0.363, 0.155) | 0.43 |
| Occupation | | | | | | |
| Fair | 0 | Reference | | 0 | Reference | |
| Good | 0.187 | (-0.102, 0.477) | 0.204 | 0.275 | (-0.034, 0.585) | 0.081 |
| Marital status | | | | | | |
| Ever married | 0 | Reference | | 0 | Reference | |
| Single | 0.264 | (0.016, 0.513) | 0.037 | 0.114 | (-0.162, 0.389) | 0.417 |
| Sexual activity | | | | | | |
| Never or none in a year | 0 | Reference | | 0 | Reference | |
| Active | 0.05 | (-0.194, 0.294) | 0.688 | -0.069 | (-0.319, 0.181) | 0.59 |
| History of HPV vaccine | | | | | | |
| No | 0 | Reference | | 0 | Reference | |
| Yes | -0.096 | (-0.402, 0.211) | 0.54 | -0.178 | (-0.500, 0.143) | 0.276 |
| Parity | | | | | | |
| > 1 | 0 | Reference | | 0 | Reference | |
| ≤ 1 | 0.196 | (-0.056, 0.448) | 0.128 | 0.009 | (-0.280, 0.297) | 0.953 |
| History of previous test results | | | | | | |
| Normal | 0 | Reference | | 0 | Reference | |
| Abnormal | -0.044 | (-0.305, 0.217) | 0.74 | 0.037 | (-0.226, 0.301) | 0.782 |
| Never had previous test | 0.304 | (0.031, 0.577) | 0.029 | 0.178 | (-0.105, 0.461) | 0.217 |
| Severity of current histopathologic findings | | | | | | |
| Normal/ inflammation | 0 | Reference | | 0 | Reference | |
| < CIN2 | 0.115 | (-0.144, 0.373) | 0.383 | 0.104 | (-0.152, 0.360) | 0.424 |
| ≥ CIN2 | 0.02 | (-0.274, 0.313) | 0.895 | -0.01 | (-0.305, 0.284) | 0.944 |

[1]Coef. (Regression coefficient) estimated by simple linear regression analysis.

[2]Coef. (Regression coefficient) by multiple linear regression analysis.

area of the country. Our participants were from the big city where health services are readily available, so not familiar to the VIA. Probably for the same reason, over 60–70% of the participants misunderstood that imaging study or tumor markers could be used as screening tests. These misconceptions might lie on the advertisement from many private hospitals in Bangkok leading to the misunderstanding.

Although 50–70% of the participants in our study had some ideas when and how frequent the screening should be, less than 20% knew that cervical cancer screening was not necessary for women after hysterectomy for non-cervical lesions and less than 40% did not know when to stop. Although the national (RTCOG) and international guidelines (USPSTF) have stated these 2 issues clearly [2, 17], they were quite specific. Knowledge of cervical cancer screening

should also include, not only how and when to start screening, but also when or condition that screening can be stopped as well.

According to knowledge about HPV, up to 20% had never heard about HPV at all. This was similar to many previous studies which reported insufficient knowledge of cervical cancer and HPV [18–24]. Specific to the women with abnormal cervical testing or lesions, one study in Sweden also reported that only 30% of patients who had high-grade CIN had good knowledge of HPV and cervical cancer [14]. Nevertheless, majority of our participants who had heard about HPV knew that HPV infection is preventable by vaccination (about 85%) and oncogenic type of HPV is the cause of disease (90.5%). This was also reported in one study from the US that 70% of healthy women had the sufficient knowledge about HPV [25]. Our Thai Gynecologic Cancer Society has been working in collaboration with the Royal Thai College of Obstetricians and Gynaecologists by launching many activities to provide knowledge to public. The health educators and professionals in areas or countries where their population still have inadequate knowledge should also focus on an introduction of HPV to the community in the national level along with the soliciting screening program. An increased knowledge and understanding of women and their communities should enhance the screening coverage up to the target.

There are several contributing factors which may impact the level of knowledge, such as, age, socio-economic status, level of education, culture, and etc. [11–13]. Previous study reported women who had never had screening were younger age (age < 45 years), pre-menopause and low monthly family income [13]. Our study found only younger age (≤ 40 year) as an independent factor associated with good knowledge. This may lie on the social environment and lifestyle of the younger generations who attach closely to media in their daily life, so more information is obtained [26].

This study had some strengths. First, this study was a survey of knowledge among women who readily had abnormal screening tests. We originally expected that these women should have better knowledge after knowing that they were having this health problem. However, the results were not as expected. This finding should be emphasized to all caregivers who could take the opportunity during the hospital visits to build up knowledge for the women themselves and their family members. Second, the questionnaire covered key points of basic and practical knowledge about cervical care screening and HPV. Furthermore, all questions were answered quite complete by the women. Specific issues which had high percentages of misconception or inadequate knowledge should be useful for health care providers to explain, discuss, or clarify with the women having medical service.

However, we were also aware of some limitations. Being a survey study via questionnaire, we did not know whether each participant had paid attention or be truthful in answering the questions about knowledge. Second, the source of health information they usually acquired was not questioned. Future study may gather this data for a genuine action plan in education planning how to deliver knowledge or health messages to the women.

In summary, many countries including Thailand have put many efforts to prevent and to reduce the prevalence of cervical cancer in response to a strong global call of WHO to eliminate cervical cancer. Aside from HPV vaccination, a good coverage of target population for cancer screening is also a key factor for the goal achievement. Because of a lack or an inadequate knowledge of cervical cancer etiology and screening, numbers of women may not adhere to health service. Hence, the policy to improve knowledge of cervical cancer and HPV is also crucial aside from the provision of health access to the target women for screening. An implementation of basic health educational program in school and/ or any other medias is needed to reinforce target women to have screening as appropriate.

## Conclusion

Despite being women at risk of having cancer due to their abnormal screening test, up to 75% of women in this study still had inadequate knowledge about cervical cancer etiology (HPV) and its screening. Younger age was the only factor associated with good knowledge. Education via various and continual electronic medias may help provide knowledge to other groups of women should be dispersed.

## Supporting information

**S1 Table. Content validity index of questionnaire about knowledge of cervical cancer screening and HPV.**
(DOCX)

**S1 Dataset.**
(XLSX)

## Acknowledgments

This work is in the umbrella project of 'Self-sampling HPV' supported by the Thai Gynecologic Oncology Society. We thank all staffs in the gynecology out-patient clinics in each participating hospitals for their administrative support.

## Author Contributions

**Data curation:** Nattapong Sreamsukcharoenchai, Supakorn Pitakkarnkul, Kristsanamon Rittiluechai, Siriwan Tangjitgamol.

**Investigation:** Uraiwan Khomphaiboonkij, Nattapong Sreamsukcharoenchai, Supakorn Pitakkarnkul.

**Methodology:** Uraiwan Khomphaiboonkij, Supakorn Pitakkarnkul, Kristsanamon Rittiluechai.

**Project administration:** Uraiwan Khomphaiboonkij.

**Supervision:** Kristsanamon Rittiluechai, Siriwan Tangjitgamol.

**Validation:** Uraiwan Khomphaiboonkij, Kristsanamon Rittiluechai, Siriwan Tangjitgamol.

**Visualization:** Uraiwan Khomphaiboonkij, Siriwan Tangjitgamol.

**Writing – original draft:** Uraiwan Khomphaiboonkij, Nattapong Sreamsukcharoenchai, Supakorn Pitakkarnkul, Kristsanamon Rittiluechai, Siriwan Tangjitgamol.

**Writing – review & editing:** Uraiwan Khomphaiboonkij, Nattapong Sreamsukcharoenchai, Supakorn Pitakkarnkul, Kristsanamon Rittiluechai, Siriwan Tangjitgamol.

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
