## [Decision Letter · Decision Letter 0]

12 Apr 2023

PONE-D-23-00077Knowledge of Thai Women in Cervical Cancer Etiology and ScreeningPLOS ONE

Dear Dr. Khomphaiboonkij,

Thank you for submitting your manuscript to PLOS ONE. After careful consideration, we feel that it has merit but does not fully meet PLOS ONE’s publication criteria as it currently stands. Therefore, we invite you to submit a revised version of the manuscript that addresses the points raised during the review process. Please submit your revised manuscript by May 27 2023 11:59PM. If you will need more time than this to complete your revisions, please reply to this message or contact the journal office at plosone@plos.org. Please include the following items when submitting your revised manuscript:A rebuttal letter that responds to each point raised by the academic editor and reviewer(s). You should upload this letter as a separate file labeled 'Response to Reviewers'.A marked-up copy of your manuscript that highlights changes made to the original version. You should upload this as a separate file labeled 'Revised Manuscript with Track Changes'.An unmarked version of your revised paper without tracked changes. You should upload this as a separate file labeled 'Manuscript'.

We look forward to receiving your revised manuscript.

Kind regards,

Trasias Mukama 

Academic Publisher 

PLOS ONE

Journal Requirements:

Reviewers' comments:

Reviewer's Responses to Questions

**Comments to the Author**

1. Is the manuscript technically sound, and do the data support the conclusions?

Reviewer #1: Yes

Reviewer #2: Yes

2. Has the statistical analysis been performed appropriately and rigorously?

Reviewer #1: Yes

Reviewer #2: No

3. Have the authors made all data underlying the findings in their manuscript fully available?

Reviewer #1: Yes

Reviewer #2: Yes

4. Is the manuscript presented in an intelligible fashion and written in standard English?

Reviewer #1: Yes

Reviewer #2: No

5. Review Comments to the Author

Reviewer #1: The manuscript is excellent. However, there is a concern ;

- In table 1, there is a question on HPV vaccination -bivalent, quadrivalent.......; my concern is do the participants have much ideas on this types because the questionnaire is self- administered one..and they have to fill by themselves. do they understand the types?

-

Reviewer #2: Comments to Authors:

The authors address an important topic regarding knowledge of the aetiology and screening for cervical cancer among high-risk Thai women. However, I would recommend the authors address the following comments

INTRODUCTION:

Line 34: Please revise statement by adding (a): In 2020, the World Health Organization (WHO) announced (a) global strategy…….

Line 36: Please revise statement by adding (that): For screening, the target is (that) 70% of women….

Line 60-63: please revise sentence for clarity.

METHODS:

General comment throughout the methods section: The authors need to provide sub-headings to structure the different sections such as study site, participants and eligibility criteria etc… This will enable the reader to keenly follow the methods undertaken by the researchers.

Line 83: Please revise statement by deleting (to): The women who 83 had prior to treatment for invasive cervical cancer…….

General comment: The authors do not mention the sampling technique they used for selection of study participants. Were all participants with abnormal cervical cytology /HPV testing included at each of the 10 participating hospitals?

Line 85-91: Authors should give a brief description of the questionnaire they used particularly for questions on cervical cancer screening (Part II) and knowledge on HPV (Part III). Was it developed by the research team or was it adopted from somewhere? If it was developed by the research team, what literature informed the questions set out? Any previous validations?

Line 92-97: The authors mention that only part 1 and 2 of the questionnaire were reviewed/revised by the expert panel of oncologists, why only these 2 parts yet the questionnaire consisted of three parts? And how many oncologists reviewed the questionnaire for this initial round? They only mention about three experts in the final round. Authors should also provide detail of the percentage agreement or Content Validity Index (CVI) among the experts at the point they reached consensus across the whole set of questions. The calculation of CVI and question evaluation of the different experts can be provided as supplementary material to the manuscript

It is good that the authors carried out reliability tests of the questionnaire. However, they should mention the reliability test statistic they used to conclude on the reliability of 0.819 and 0.907.

Line 113-115: Authors mention that they compared “good knowledge” and “not good knowledge” scores using Chi-square or student t-tests but these comparisons do not appear anywhere in the results section. I recommend authors remove this statement.

Line 115-116: The authors should mention the level of significance/cut off used for significance for the analyses conducted.

Results

Line 118-119: please revise sentence for grammar and clarity

Line 121-122: Revise accordingly by adding (were) or rephrase for correct sentence structure: More than half of the participants (were) aged < 40 years old.

Table 5: There is discordance in the methods and results sections regarding data analysis technique used by the authors. In the methods section, authors mention that they applied logistic regression for multivariable analyses; however, this appears not to be the case in Table 5. Authors should correct this in the methods section.

Table 5: The study should elaborate what variable selection method they used for the multivariable model. It appears that they carried all covariates to the multivariable model even those not adding any value to the model. Was the final model tested for goodness of fit or what criteria was applied to select the final model?

DISCUSSION:

General comment throughout the discussion section: Authors should correct some of the grammatical errors/ poorly structured sentences like on line: 188-189, 190-192, 199, 208-209.

Line 206: The explanation of the 32% of the women being familiar with VIA does not come out clearly. If health services are readily available, how does this justify the low awareness of women on this particular item? I would think this should be the other way round. I recommend authors provide more justifying evidence

Line 220-224: Authors compare their findings among healthy women where 70-77% were unknowledgeable about HPV in China and Macau. There is no similarity in these findings i.e. 20% vs 70%-77%, meaning healthy women are way less knowledgeable on HPV as cause of Cervical cancer compared to the current study which recruited women with abnormal cytology. I recommend authors to find other relevant papers among women with similar characteristics for appropriate comparisons.

6. PLOS authors have the option to publish the peer review history of their article (what does this mean?). If published, this will include your full peer review and any attached files.

Reviewer #1: No

Reviewer #2: No

---

## [Author Response · Author response to Decision Letter 0]

27 Apr 2023

Reviewer #1: The manuscript is excellent. However, there is a concern:

Thank you for your kind words. 

1. In table 1, there is a question on HPV vaccination -bivalent, quadrivalent.......; 

my concern is do the participants have much ideas on these types because the questionnaire is self- administered one and they have to fill by themselves. do they understand the types?

Re: HPV vaccination had been launched in our country for quite a long period. It has also been included in the national immunization program for the girls at certain ages, so we included this in our questionnaire. 

However, we had a choice of “don’t know” for those who had but could not recall or knew the type of vaccine received (n= 7). 

Reviewer #2: Comments to Authors:

The authors address an important topic regarding knowledge of the aetiology and screening for cervical cancer among high-risk Thai women. However, I would recommend the authors address the following comments

Thank you for your thorough assessment and comments. We have addressed each of your question or comment as follow.

INTRODUCTION

1. Line 34: Please revise statement by adding (a): In 2020, the World Health Organization (WHO) announced (a) global strategy.

Re: We added a letter ‘a’. (line 39)

2. Line 36: Please revise statement by adding (that): For screening, the target is (that) 70% of women.

Re: We added the word ‘that’. (line 41)

3. Line 60-63: please revise sentence for clarity.

Re: Revision was made (lines 65-69).

METHODS: General comment throughout the methods section: 

4. The authors need to provide sub-headings to structure the different sections such as study site, participants, and eligibility criteria etc… This will enable the reader to keenly follow the methods undertaken by the researchers. 

Re: Per your suggestion, we revised this Methods section after general information of the study by adding sub-headings of: Participants and eligibility criteria (lines 85), Study questionnaires (lines 92), Data collection (lines 111), and Statistical analysis (lines 118).

5. Line 83: Please revise statement by deleting (to): The women who 83 had prior to treatment for invasive cervical cancer.

Re: The word ‘to’ was deleted.(lines 89)

6. General comment: The authors do not mention the sampling technique they used for selection of study participants. Were all participants with abnormal cervical cytology /HPV testing included at each of the 10 participating hospitals?

Re: All who consented to participate, 50 from each hospital. We added these data under the Participants and eligibility criteria (lines 90-91).

7. Line 85-91: 

• Authors should give a brief description of the questionnaire they used particularly for questions on cervical cancer screening (Part II) and knowledge on HPV (Part III). 

Re: We added details of the questions/ statements in the questionnaire pf Part II and Part III (lines 96-100).

• Was it developed by the research team or was it adopted from somewhere? If it was developed by the research team, what literature informed the questions set out? Any previous validations? 

Re: The questionnaire was developed by the researchers by reviewing relevant publications particularly in our country (including our previous works: ref# 12-13) which reported the problems for not having cervical cancer screening. Each item was discussed extensively in our group before revision and validation. 

8. Line 92-97: 

• The authors mention that only part 1 and 2 of the questionnaire were reviewed/revised by the expert panel of oncologists, why only these 2 parts yet the questionnaire consisted of three parts? 

Re: Apologize for our mistake. It was Part II and Part III which were tested for validity and reliability of the questionnaire. Part I involved only personal data.

• And how many oncologists reviewed the questionnaire for this initial round? They only mention about three experts in the final round. 

Re: All 20 researchers (gynecologic oncologists) contemplated, discussed, and revised the contents of the questionnaire until consensus (added as lines 102-104).

• Authors should also provide details of the percentage agreement or Content Validity Index (CVI) among the experts at the point they reached consensus across the whole set of questions. The calculation of CVI and question evaluation of the different experts can be provided as supplementary material to the manuscript.

Re: We had not primarily presented the CVI because the values were 1 due to the simplicity of the questions. Per your advice, we added the method to obtain the CVI (lines 104-108) and presented it as Supplementary Table A. 

9. It is good that the authors carried out reliability tests of the questionnaire. However, they should mention the reliability test statistic they used to conclude on the reliability of 0.819 and 0.907. 

 Re: We added the reliability test with Cronbach’ alpha coefficients (lines 108-110)

10. Line 113-115: Authors mention that they compared “good knowledge” and “not good knowledge” scores using Chi-square or student t-tests, but these comparisons do not appear anywhere in the results section. I recommend authors remove this statement. 

Re: Apologize for our mistake. After a thorough review that we presented the knowledge comparison of the women with continuous data of score, we removed the sentence of binary group comparison per your knowledgeable recommendation. 

11. Line 115-116: The authors should mention the level of significance/cut off used for significance for the analyses conducted. 

Re: We added the level of significance (line 135-136)

Results

12. Line 118-119: please revise sentence for grammar and clarity.

Re: We revised the sentence to clarify it (lines 138-141).

13. Line 121-122: Revise accordingly by adding (were) or rephrase for correct sentence structure: More than half of the participants (were) aged < 40 years old. 

Re: Revised (line 143)

14. Table 5: There is discordance in the methods and results sections regarding data analysis technique used by the authors. In the methods section, authors mention that they applied logistic regression for multivariable analyses; however, this appears not to be the case in Table 5. Authors should correct this in the methods section. 

Re: We reviewed and corrected the description in the Method section. 

15. Table 5: The study should elaborate what variable selection method they used for the multivariable model. It appears that they carried all covariates to the multivariable model even those not adding any value to the model. Was the final model tested for goodness of fit or what criteria was applied to select the final model?

Re: We had designed our questionnaire to collect any possible features of the women that may impact knowledge of cervical cancer. So, we used the exploratory method for the multivariable analysis by including all potential factors to serve the purpose.

DISCUSSION:

16. General comment throughout the discussion section: Authors should correct some of the grammatical errors/ poorly structured sentences like on line: 188-189, 190-192, 199, 208-209. 

Re: We revised the sentence to clarify them accordingly (lines 211-221, 228-229).

17. Line 206: The explanation of the 32% of the women being familiar with VIA does not come out clearly. If health services are readily available, how does this justify the low awareness of women on this particular item? I would think this should be the other way round. I recommend authors provide more justifying evidence. 

Re: We expanded the explanation of this issue that VIA is promoted in low-resource areas with a deficit of physicians (gynecologist and/or pathologist). Our participants lived in the big cities where all medical services are readily available, so they were not acquainted to VIA (lines 237-239).

18. Line 220-224: Authors compare their findings among healthy women where 70-77% were unknowledgeable about HPV in China and Macau. There is no similarity in these findings i.e. 20% vs 70%-77%, meaning healthy women are way less knowledgeable on HPV as cause of Cervical cancer compared to the current study which recruited women with abnormal cytology. I recommend authors to find other relevant papers among women with similar characteristics for appropriate comparisons. 

Re: Per your advice, we re-reviewed and found one publication in PLoS One--- focusing on knowledge of women with CIN2+. The article was added as ref# 14 (lines 252-254).

---

## [Editor Report · Decision Letter 1]

7 May 2023

Knowledge of Thai Women in Cervical Cancer Etiology and Screening

PONE-D-23-00077R1

Dear Dr. Khomphaiboonkij,

We’re pleased to inform you that your manuscript has been judged scientifically suitable for publication and will be formally accepted for publication once it meets all outstanding technical requirements.

Kind regards,

Dr. Trasias Mukama

Academic Editor

PLOS ONE

---

## [Editor Report · Acceptance letter]

10 May 2023

PONE-D-23-00077R1 

Knowledge of Thai Women in Cervical Cancer Etiology and Screening 

Dear Dr. Khomphaiboonkij:

I'm pleased to inform you that your manuscript has been deemed suitable for publication in PLOS ONE. Congratulations! Your manuscript is now with our production department. 

Kind regards, 

on behalf of

Dr. Trasias Mukama 

Academic Editor

PLOS ONE